# How wear, age, and sex relate to enamel chipping in Cayo Santiago rhesus macaques (*Macaca mulatta*)

**Debbie Guatelli-Steinberg**[1]*, **Luke D. Fannin**[2], **Qian Wang**[3]

**1** Department of Anthropology, The Ohio State University, Columbus, Ohio, United States of America,
**2** Department of Anthropology, Dartmouth College, Hannover, New Hampshire, United States of America,
**3** Department of Biomedical Sciences, Texas A&M University College of Dentistry, Dallas, Texas, United States of America

* guatelli-steinberg.1@osu.edu

## Abstract

In living primates, bite forces required to fracture hard-object foods can result in high frequencies of chipped teeth, providing a comparative basis for inferring hard-object feeding in the fossil record. Yet, inferring hard-object feeding from chipping frequencies is complicated by factors such as dental wear, age, and sex that might also affect them. Using dental remains of rhesus macaques (*Macaca mulatta*) from Cayo Santiago, Puerto Rico, we investigated whether these factors covaried with molar chipping frequencies. We hypothesized that dental wear, because it has a direct relationship to chip formation, would be a stronger predictor of chipping frequencies than age. We also hypothesized that if a sex difference were found, males would have higher frequencies of chipping than females, which is the most common sex difference in chipping found in humans. Samples consisted of 36–38 molars from females and 60–63 molars from males per molar type. Dental wear was measured and chips were identified on consistently oriented occlusal surface photographs. We performed a logistic General Linear Mixed Model (GLMM) of chipping with data on wear and chipping for the six different molar types treated as repeated measures for individuals. The GLMM revealed statistically significant effects for wear and sex, but not for age, on chipping. Our results suggest that wear has a greater effect on chipping frequencies than age, and that sex, at least in this sample, can also affect chipping frequencies. Sex differences in chipping frequencies of the magnitude found here could potentially obscure dietary signals inferred from chipping in studies of fossil primates. These findings suggest that analyzing chipping frequencies with respect to wear and sex could help improve the accuracy of dietary reconstruction of fossil primate diets based on chipping.

provided the original author and source are credited.

**Data availability statement:** The data have been uploaded along with the paper in a supplementary file.

**Funding:** This project was supported by National Science Foundation grants to D.G.-S. and Q.W. (NSF RIDIR 1926528, 1926601). URL: https://www.nsf.gov/ NSF had no role in the study design, data collection and analysis, decision to publish, or preparation of the manuscript.

**Competing interests:** The authors have declared that no competing interests exist.

## Introduction

Analysis of dental chipping in fossil teeth [1–10] is one of several approaches for reconstructing diet in fossil primates, including hominins [11,12]. Biting objects with the force necessary to fracture them can cause enamel, a brittle material, to crack, in some instances creating a chip (or more precisely a chipping scar) on the enamel surface [1]. The perimeters of chips formed during an individual's lifetime are smoothed by continued tooth use, making it possible to distinguish antemortem from postmortem chipping [1,13–15]. High frequencies of antemortem chips found in extant primates that feed on hard-objects provide the comparative basis for inferring hard-object feeding in fossil primates [16–19].

Direct indicators of diet, such as enamel chipping [1–10], dental microwear [20–30], and the isotopic composition of enamel [29–36], are essential for interpreting the functional craniodental anatomy of fossil primates and reconstructing their ecologies [11–12]. For example, low frequencies of chipping [5], smooth microwear textures [25,28], and high ratios of carbon 13 to carbon 12 (indicating consumption of $C_4$ resources [34]), all suggest that the hominin *Paranthropus boisei* was not a hard-object feeder despite its thickly enameled teeth and heavily buttressed craniofacial skeleton.

While a useful indicator of hard food comminution in fossil species, chipping frequencies are likely to be affected by factors other than diet, most notably dental wear and age [11,36]. Understanding to what extent these factors can affect chipping frequencies is necessary for improving the accuracy of dietary inferences that are drawn from them. To this end, the first objective of this study is to assess to what extent chipping frequencies in a living primate model are related to tooth wear and age.

Although previous studies have found relationships between chipping and tooth wear [2: *Homo naledi*, 17: *Cercocebus atys*] as well as age [37]: modern human archaeological samples], to our knowledge no study has yet examined chipping frequencies in living primates in relation to both wear and age to assess the relative contribution of these factors to chipping frequencies. The Cayo Santiago rhesus macaque sample, consisting of dental remains from known-age individuals, makes it possible to investigate associations between chipping frequencies and both dental wear and age. In addition, since the Cayo Santiago rhesus monkey remains are also of known sex, it is possible to assess whether sex and chipping frequencies covary, which is the second objective of the present study. Sex as a factor that might be relevant for understanding variation in chipping frequencies is rarely considered in the chipping literature on fossil hominins and primates, likely because determining the sex of skeletal remains is challenging, though this is now possible through the analysis of AMELX and AMELY proteins in teeth [38–42].

As a model for understanding non-food factors affecting enamel chipping in fossil primates (including hominins), rhesus monkeys have two characteristics that should be borne in mind. First, their relative enamel thickness (RET) ranges from 12.16 for the UM1 to 14.10 for the LM3 [43], placing them into Martin's "intermediate/thin" category for primate enamel thickness [44]. Thick enamel, such as that found in early

fossil hominins [45], could be expected to make their teeth less likely to chip than those of rhesus monkeys. In addition, *Macaca mulatta* exhibits a small degree of molar sexual dimorphism, with males having molar dimensions ranging from being comparable to those of females (e.g., posterior bucco-lingual average for the upper M3 of 6.5 mm for both males and females) to a maximum difference of 0.9 mm (for the lower M3 mesio-distal dimension) [46]. Thus, with respect to using rhesus macaques as a model for understanding how the effects of dental wear may affect chipping in fossil primates, we note that rhesus macaque enamel falls into an intermediate range for primate enamel thicknesses and is thinner than that of early hominins [45]. With respect to understanding associations between sex and chipping, it is worth noting that rhesus macaques do not have particularly sexually dimorphic molars.

Dental wear is theorized to increase the probability of chipping [10,47–49]. Chips form when enamel cracks (i.e., fractures within the enamel) travel from the loading point downward along the loading axis and veer towards the closest molar sidewall [47]. When they reach the sidewall, these fractures cause the enamel to spall. In worn molars, as compared to unworn molars, the distance between a downward-travelling crack and the molar sidewall is smaller because the sidewall is less inclined. The smaller distance of the crack to the molar sidewall decreases the critical load for chipping [47–49] and reduces absolute chip size [16]. Thus, there is a direct theoretical link between dental wear and the chance of a chip forming at the edge of a tooth.

Given these theoretical expectations, wear is normally expected to make molars more likely to chip. Yet another plausible scenario is that wear might instead erase small chips from enamel surfaces prior to death [11,36]. Under this scenario, chipping frequencies obtained from dental remains would underestimate the frequency of chipping in the living. It seems, however, that the overall effect of wear is to increase rather than decrease chipping frequencies: in both *C. atys* [17,19] and *H. naledi* [2] worn teeth were found to have higher frequencies of chipping than unworn teeth.

Age is also thought to increase the probability of chipping. First, the time available for chipping to occur is greater in animals that live longer lives [11,36]. Second, enamel becomes more brittle with age, causing it to become less fracture resistant [50]. The brittleness of older enamel is caused by a reduction in the interprismatic protein matrix [51] and a concomitant increase in mineral density [52]. In humans, studies have found enamel cracks to be more prevalent in older individuals [53–55]. With respect to enamel chipping specifically, Turner & Cadien [37] found enamel chips to be more common in older age groups of prehistoric Arctic peoples. Yet, because there is usually a high correlation between dental wear and age, forming the basis for using dental wear as an aging technique [56–59], it is not clear if previous studies linking age to chipping [37] or wear to chipping [2,17] have more to do with wear or age. Because theoretical models demonstrate a direct relationship between dental wear and a specific susceptibility to edge-chipping [47–49], we hypothesized that there would be more evidence linking chipping to wear than to age in the Cayo Santiago rhesus macaque dental sample.

On theoretical grounds, it is not clear whether to expect a sex difference in enamel chipping frequencies. Sex differences in chipping frequencies have been found in ancient human samples (summarized in [15]). Males generally have higher frequencies than females (Quadrella and Vincenne-Campocchiaro, Italy [14]; Taforalt, Morocco [60]; Saint Lawrence Island Inuit and Medieval Norwegians [15]), although in only one of these cases (Saint Lawrence Island Inuit [15]) was the sex difference in chipping statistically significant. The causes of this sex difference are unknown, though they could have to do with a number of factors: sex differences in enamel thickness, diet, bite force, and/or tooth grinding. Based on these empirical findings in humans, we hypothesized that sex might be a statistically significant predictor of chipping frequencies. We predicted that, if a sex difference were found, then like human males, Cayo Santiago males would have significantly higher chipping frequencies than females.

In the Cayo Santiago rhesus monkey population, it is not likely that a sex difference in chipping frequency, if found, would be related to sex differences in food material properties, as both sexes are provisioned with identical monkey chow and forage on naturally growing plants [61]. It is worth noting that the majority of Cayo Santiago rhesus monkeys are also geophagic, with females generally consuming more soil than males [62]. Terrestrial primate species tend to have higher

chipping prevalence than non-terrestrial ones, suggesting that grit may play a role in chip formation [18]. It is therefore possible that the sex difference in geophagic behavior might lead to higher chipping frequencies in females than males. In the present study, we also include a comparison of monkeys that spent their whole lives ranging free on the island to those who were transferred into captivity to the facility at Sabana Seca, where they presumably would have had less opportunity to consume soil. By including a comparison of chipping in the free-ranging vs. captive monkeys, we have an opportunity to assess whether geophagy has an influence on chipping frequencies as might be expected if abrasion affects them. It is also worth noting that Cayo Santiago monkeys have been observed gnawing on tree limbs and rocks [63], though it is not clear if there is a sex difference in this behavior.

We further explored sex differences in enamel chipping by comparing male and female occlusal areas. It has been suggested that larger occlusal areas provide more space for chipping than smaller occlusal areas [11]. This suggestion was given in the context of interpreting inter-specific differences in chipping frequencies but could potentially explain sex differences in chipping within a species if males have larger occlusal areas than females. Male primates generally have larger teeth (including molars) than females [46]. Although a finding of larger occlusal areas in males in this study is therefore expected, we evaluate whether sex differences in occlusal areas for all molar types are the same, of if they vary by molar type. Molar types with larger sex differences in occlusal area might be expected to exhibit greater sex differences in chipping frequency if molar size influences chipping frequencies.

We also compared males and female edge chip size to evaluate if there is any evidence of bite force differences between the sexes. Chai & Lawn [64] showed that dimension "h" of scallop-shaped edge chips increased with increasing load. In molars, "h" is the distance between the presumed loading point and the molar's sidewall (see Fig 1, this study). This distance is correlated with bite force across primates [1]. Because adult rhesus macaque males can exert higher bite forces than adult females [65], we explored whether this might be reflected in their chip sizes. The use of sensory information to modulate bite forces during chewing [66,67], on the other hand, suggests that animals bite only as hard as necessary to fracture foods [17]. If so, then given the similar diets of male and female Cayo Santiago rhesus monkeys, sex differences in chip size would not be expected. Unfortunately, we could not adequately test predictions about chip size owing to sample size limitations—there were too few scallop-shaped edge chips with clearly defined boundaries that could be accurately measured for meaningful statistical analysis. We report these here as exploratory measurements, commenting on any apparent trends.

## Materials and methods

### Ethics statement

This research adhered to the American Society of Primatologists Principles for the Ethical Treatment of Non-human Primates. The research reported here was noninvasive. It was conducted on dental remains using non-destructive methods. We had permission to use a census database and to collect data on skeletons from CPRC, per a "General Collaboration Agreement" between Qian Wang and the Caribbean Primate Research Center (CPRC). Our interdisciplinary team led by Qian Wang, funded by NSF (RIDIR 1926402, 1926481, 1926528, and 1926601), aimed at integrating related data of rhesus macaques at Cayo Santiago for studies of growth and development, acclimation and adaptation, and health and diseases as a non-human primate model. Raw data for this study are included in the SI as Excel files. Access to the raw census database can only be granted by CPRC. The authors declare no conflict of interest.

### Study sample and measurements

The dental remains used in this study are those of rhesus macaques (*Macaca mulatta*) that lived on the island of Cayo Santiago, lying one kilometer from the coast of Puerto Rico [68,69]. Primatologist Clarence Carpenter transported 409 rhesus macaques (including 40 adult males and 183 adult females) from India to the island in 1938, establishing the

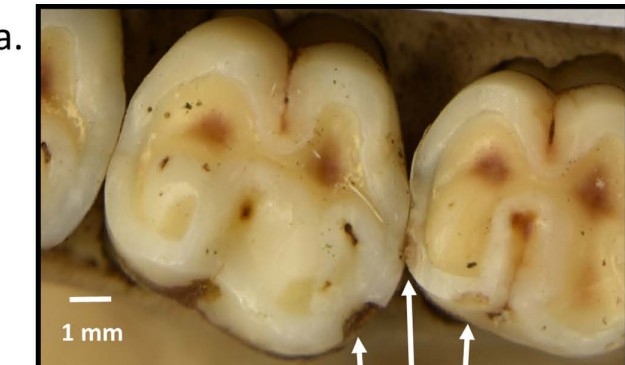

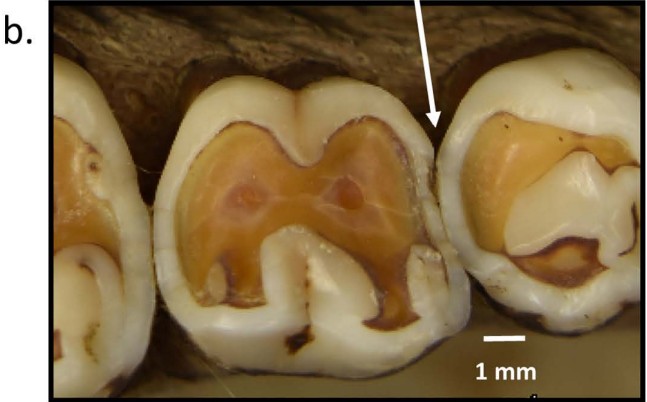

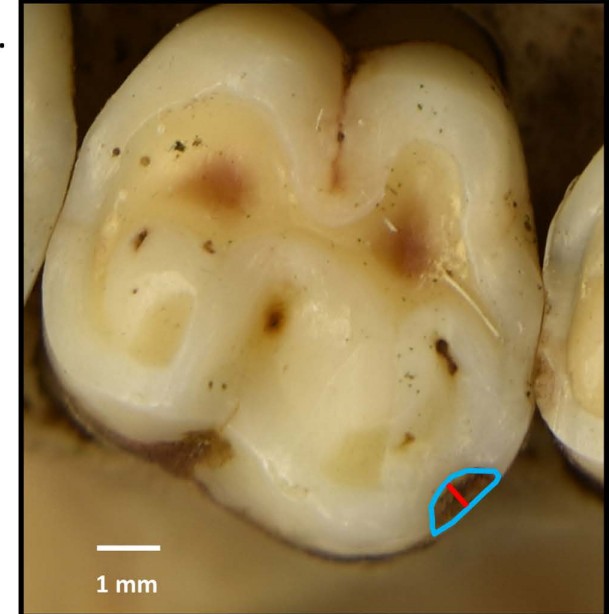

**Fig 1. Examples of enamel chips and chip measurements.** a: Chip on upper left M1 (on right) and M2 (on left) of specimen 2962. Arrows indicate chips. b: Chip on lower left molar of specimen 2983, indicated by an arrow. c: How scallop-shaped edge chips were measured. Red line represents "h" (see text for definition); area outlined in blue is the chip area on the lower left M2 of specimen 2962.

colony. Their descendants live on the island today. Donald Sade of the Caribbean Primate Research Center (CPRC) at the University of Puerto Rico Medical campus began a skeletal collection of the monkeys' remains in 1970. In the 1980's, Jean E. Turnquist and Matthew J. Kessler initiated a regular program of carcass collection. The CPRC Museum, now called the Laboratory of Primate Morphology, was established in 1982 to house the monkeys' skeletal remains.

The sample of teeth used in this study derive from individuals included in the 1985 "round-up", conducted for the purpose of immunizing the colony against tetanus [63]. At this time, the monkeys were physically examined and scored for dental wear. In a recent study dental wear was scored on the dental remains of a subset of the monkeys included in the round-up, making it possible to examine dental wear at two points in these monkey's lives—one at the time of the round up and the other at death [70]. The sample and dental wear measurements used for that study are the same as those used here.

Teeth were photographed with a Nikon D5600 digital camera mounted on a tripod. A bubble level was used to orient the camera's optical axis orthogonally to the occlusal surfaces of maxillary and mandibular molars [70]. Mandibles and maxillae were positioned with clay or in a container of uncooked rice grains so that their occlusal surfaces appeared level (Fig 2). Photographs were taken with a scale used to perform measurements in ImageJ [71].

To quantify wear, the polygon tool in ImageJ was used to measure areas of dentine exposure. These areas were summed, divided by the molar's occlusal area (Fig 2), and multiplied by 100 to obtain the percentage of the molar's

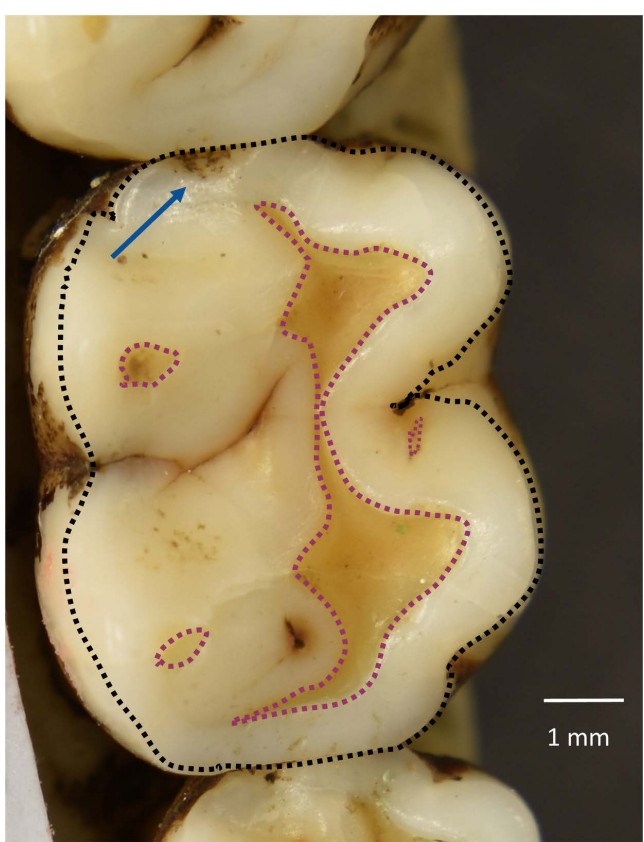

**Fig 2. Measurement of wear on molar occlusal surfaces.** Figure showing how worn areas of exposed dentine (yellow patches outlined with dotted magenta lines) as well as occlusal areas (surface of molar outlined with dotted black line) were measured. Tooth pictured is the lower left M2 of specimen 1218, which also has an enamel chip, indicated by the blue arrow.

occlusal surface that was worn (as in [72,73]). One molar was measured per antimeric pair, based on whether the left or right antimere was best oriented. If both antimeres were well-oriented, then choice of right or left molar was alternated between each pair of antimeres. All upper and lower molars were analyzed for antemortem enamel chipping using previously published methods [17,19] and chips were scored as antemortem if fractures exhibited smoothed or blunted edges. Examples of enamel chips are shown in Fig 1. We restricted our analysis to teeth with fully-intact crowns, as postmortem alteration can remove potential chips, thereby deflating total chipping frequencies. For our statistical analyses, for each antimeric pair, we used the most-well oriented antimere for both wear and chipping.

The sex and age of individuals were incorporated into the analysis of chipping frequences. Exact age is known for these individuals, whose dates of birth and death were recorded. Also included in our analysis of chipping frequencies was the year of birth and the length of time any monkeys transferred to the captive facility at Sabana Seca, Puerto Rico were housed there. Year of birth was included as a covariate because our sample spanned a wide range of birth years – from 1957 to 1979. Time in captivity was included as a covariate because while in the captive facility monkeys do not have access to the wild plants they consume on the Island of Puerto Rico (approximately 49.8% of their diets [61]) and there is minimal opportunity for geophagy.

For the analysis of chip dimensions, both "h" and the area of the chip were measured in ImageJ (see Fig 1). As Chai noted [10] there are various factors that limit the ability to measure chips from images of occlusal surfaces: tooth decay, dirt, wear, and slight variations in specimen orientation among them. To this we add that because antemortem chips have smooth boundaries, it is sometimes difficult to delineate the boundaries of chips clearly enough to confidently measure them. In addition, we measured only chips that had a "scallop-shape" so that we could apply the model linking them to bite force (1,10,64]. By limiting our measurements to scallop-shaped chips with clear boundaries, we had only a small sample of chips to measure for male-female chip size comparisons. Both "h" and chip occlusal areas were scaled to the occlusal area of the crown, as the relationship between bite force and "h" across species was found for "h" scaled to crown size [1].

## Statistical analysis

Chipping was analyzed as a binary variable, with "0" representing no chips on a molar and "1" representing one or more chips on a molar. Basic descriptive statistics were calculated by sex for chipping frequencies, dental wear, age-at-death and molar occlusal areas.

We examined correlations between wear and age at death to allow us to develop a plan for statistical analysis. To examine these correlations, we first assessed the normality of these variables. For age-at-death, the data were normal (Shapiro-Wilk statistic 0.989, $p = 0.591$. For wear, for all tooth types, the data were not normal (Shapiro-Wilk statistic $= 0.953$, $p = 0.000$ for UM1; Shapiro-Wilk statistic $= 0.909$, $p = 0.909$ for UM2; Shapiro-Wilk statistic $= 0.878$, $p = 0.000$ for UM3; Shapiro-Wilk statistic $= 0.944$, p $= 0.000$ for LM1; Shapiro-Wilk statistic $= 0.920$, $p = 0.000$ for LM2; Shapiro-Wilk statistic $= 0.854$, $p = 0.000$ for LM3). Because wear data were not normal, correlations (reported under Results) were calculated using the non-parametric Spearman's Rank test.

Because the correlations we found were moderate to high (see Results), we chose to analyze the data in two ways: first in a General Linear Mixed Model (GLMM) regression analysis at the level of the individual, reported here in the main text, and second in a series of logistic regression analyses at the level of tooth type, included in the Supplementary Information file (entitled S1 Text: Supplementary Information: Analysis by Tooth Type). GLMM models are robust to collinearity of predictors, such as the expected correlation between age-at-death and wear in our data set [74]. Shielzeth and co-authors [74] found that weak to moderate correlations had little effect on the precisions of GLMM estimates but that there were large effects on the precision of estimates for predictors with correlations as high as 0.8. Correlations for wear and age in our data set do not reach this level but for some tooth types, they are close (see Results). For this reason, the by-tooth level of analysis we include in the SI involves a series of separate regressions for wear and age-at-death. Separating these two variables in by-tooth analysis reported in the SI removes any concern about the collinearity of wear and

age-at-death, providing a check on our primary results from the GLMM. Furthermore, performing these supplementary analyses (fully described in the SI) at the level of tooth type makes it possible for researchers interested in potential tooth type effects to examine how predictors relate to chipping at the level of tooth type.

Our GLMM analysis was performed in SAS 9.4 using PROC GLMMIX. A binary distribution and a logit link function was used for logistic regression. The individual (specimen) was treated as a random variable and tooth types were treated as fixed repeated measures. We designated individual as random because we believe that factors affecting chipping are random with respect to individuals. We treated tooth type as a fixed effect because we expected that molar type would have a consistent effect on chipping. For example, third molars erupt later than second molars and tend to have thicker enamel than first and second molars, at least in humans [75]. Both factors would be expected *a priori* to result in lower frequencies of chipping for third molars relative to first and second molars. In our GLMM, we analyzed chipping as a function of the following predictors: wear (percent of occlusal surface that is worn), age-at-death, tooth type, sex, year-of-birth, and time spent in the captive facility at Sabana Seca.

To further explore sex differences in chipping in our sample, we compared males and females in terms of occlusal areas using a repeated measures analysis in SAS 9.4 with Proc Mixed. This procedure allows missing data (for some individuals one or more molar types were missing) and takes the correlated nature of repeated measures on the same individual into account. Here, we regressed occlusal areas on sex, tooth type, and their interaction. Although sex differences in molar size were expected, we quantified the difference and evaluated whether molars with the largest sex difference in occlusal area evinced the greatest sex difference in chipping frequencies.

Finally, to compare males and females in terms of chip size, we report the basic statistics for "h," "chip area," and both values scaled to occlusal area. Sample sizes were too small for a robust statistical analysis. We comment on whether there appears to be a male vs. female bias in chip sizes, but caution that this is a preliminary analysis that requires a larger sample for statistical testing.

## Results

Table 1 gives the descriptive statistics for chipping frequencies by sex, percentage of molar occlusal surfaces that were worn, age-at-death, and occlusal area of the crown. Note that numerically, males have higher chipping frequencies than females, that male and female molars in this sample are comparable in terms of wear, that females are on average three years older than males in our sample, and that male occlusal areas are greater than those of females.

Box plots comparing individuals with or without enamel chips with respect to age and wear are given for each upper molar type in Fig 3. Similarly, box plots of individuals with or without enamel chips with respect to age and wear are given for each lower molar type in Fig 4.

Correlations between the percentage of molar occlusal surface that is worn and age-at-death range from a low of 0.321 for the upper third molar to a high of 0.763 for the lower second molar. All correlations for the six molar types are statistically significant at the 0.001 level (Table 2). These results confirmed the expectation of moderate to high correlations between age and wear that affected our decision to analyze the data in two ways: first in a GLMM analysis at the level of the individual reported here in the main text and second in a series of logistic regression analyses at the level of tooth type, included in the Supplementary Information file (entitled Supplementary Information: Analysis by Tooth Type).

The GLMM analysis results are presented in Table 3 for 97 individuals (some individuals were eliminated by the GLIMMIX procedure because of missing information). Wear and sex are the only two statistically significant predictors (Type III fixed effect for wear: F-value = 4.30, $p = 0.0387$; Type III fixed effect for sex: 7.15, $p = 0.0089$.). The GLMMIX procedure modeled the probability that chipping is zero. For wear, the regression coefficient estimate was −0.01868 ($p = 0.0387$), meaning that as wear increases, the probability of not having a chip decreases. For sex, the regression coefficient for Female vs. Male was 0.7583 vs. 0 ($p = 0.0089$), meaning that being female increased the probability of not having a chip.

**Table 1. Descriptive statistics for age-at-death for individuals; percentage of occlusal surface that is worn, chipping frequencies, and occlusal area (by sex and tooth type).**

| Maximum number of Individuals | Number of Females | Number of Males | Females: Mean Age at Death in Years (SD) | Males: Mean Age at Death (SD) |
|---|---|---|---|---|
| 101 | 38 | 63 | 19.4 (5.3) | 16.4 (5.6) |
| Molar Type | Number of Females | Number of Males | % Females with Chipped Molars | % Males with Chipped Molars |
| UM1 | 36 | 63 | 19.4 | 26.9 |
| UM2 | 38 | 63 | 10.5 | 31.8 |
| UM3 | 37 | 63 | 8.1 | 11.1 |
| LM1 | 38 | 62 | 15.8 | 27.4 |
| LM2 | 36 | 61 | 13.9 | 14.8 |
| LM3 | 37 | 60 | 8.1 | 25.0 |
| Molar Type | Number of Females | Number of Males | Females: Mean (SD) percentage of occlusal surface that is worn | Males: Mean (SD) for percentage of occlusal surface that is worn |
| UM1 | 38 | 62 | 36.6 (17.3) | 36.4 (19.1) |
| UM2 | 38 | 63 | 24.9 (13.7) | 25.3 (15.9) |
| UM3 | 37 | 63 | 18.2 (11.3) | 17.7 (10.0) |
| LM1 | 38 | 62 | 40.5 (19.8) | 40.2 (19.3) |
| LM2 | 36 | 61 | 28.7 (18.0) | 28.4 (18.1) |
| LM3 | 37 | 60 | 14.7 (12.4) | 14.6 (11.3) |
| Dentition | Number of Females | Number of Males | Females: Mean Age at Death in Years (SD) | Males: Mean Age at Death (SD) |
| Upper | 37 | 63 | 19.4 (5.3) | 16.4 (5.6) |
| Lower | 38 | 63 | 19.4 (5.3) | 16.4 (5.6) |
| Molar Type | Number of Females | Number of Males | Female Occlusal Areas (mm²) (SD) | Male Occlusal Areas (mm²) (SD) |
| UM1 | 38 | 62 | 37.6 (4.7) | 38.2 (3.9) |
| UM2 | 38 | 63 | 48.7 (6.3) | 49.9 (6.0) |
| UM3 | 37 | 63 | 42.1 (6.5) | 45.4 (5.2) |
| LM1 | 38 | 61 | 29.0 (2.5) | 31.3 (3.3) |
| LM2 | 38 | 63 | 41.2 (4.0) | 44.1 (4.8) |
| LM3 | 37 | 62 | 47.9 (5.3) | 51.4 (5.1) |

Tooth type, age-at-death, year-of-birth, and time spent in Sabana Seca were not statistically significant predictors (see Table 2 for values), although year-of-birth was close to the alpha level of 0.05 (F-value 3.75; $p = 0.0559$).

The repeated measures regression of occlusal area on sex revealed statistically significant effects for sex and molar type, as well as their interaction (Table 4). The interaction effect is apparent in Fig 5: there is very little difference in average molar size for the UM1 and UM2, with larger differences evident for the UM3, LM1, LM2, and LM3. This is noteworthy because UM2 is one of the molar types in which a sex difference in chipping frequency was found, with the other being the LM3.

Comparison of "h" and chip areas for males and females both with and without scaling to occlusal area (Table 5) do not show a clear pattern of male greater than female chip sizes. Average chip dimensions are greater in males for some molar types but not for others.

## Discussion

In this study of chipping in Cayo Santiago rhesus macaques, we hypothesized that wear would be more closely related to chipping frequencies than age. Theoretical models [47–49] link wear to enamel chipping specifically, while age is

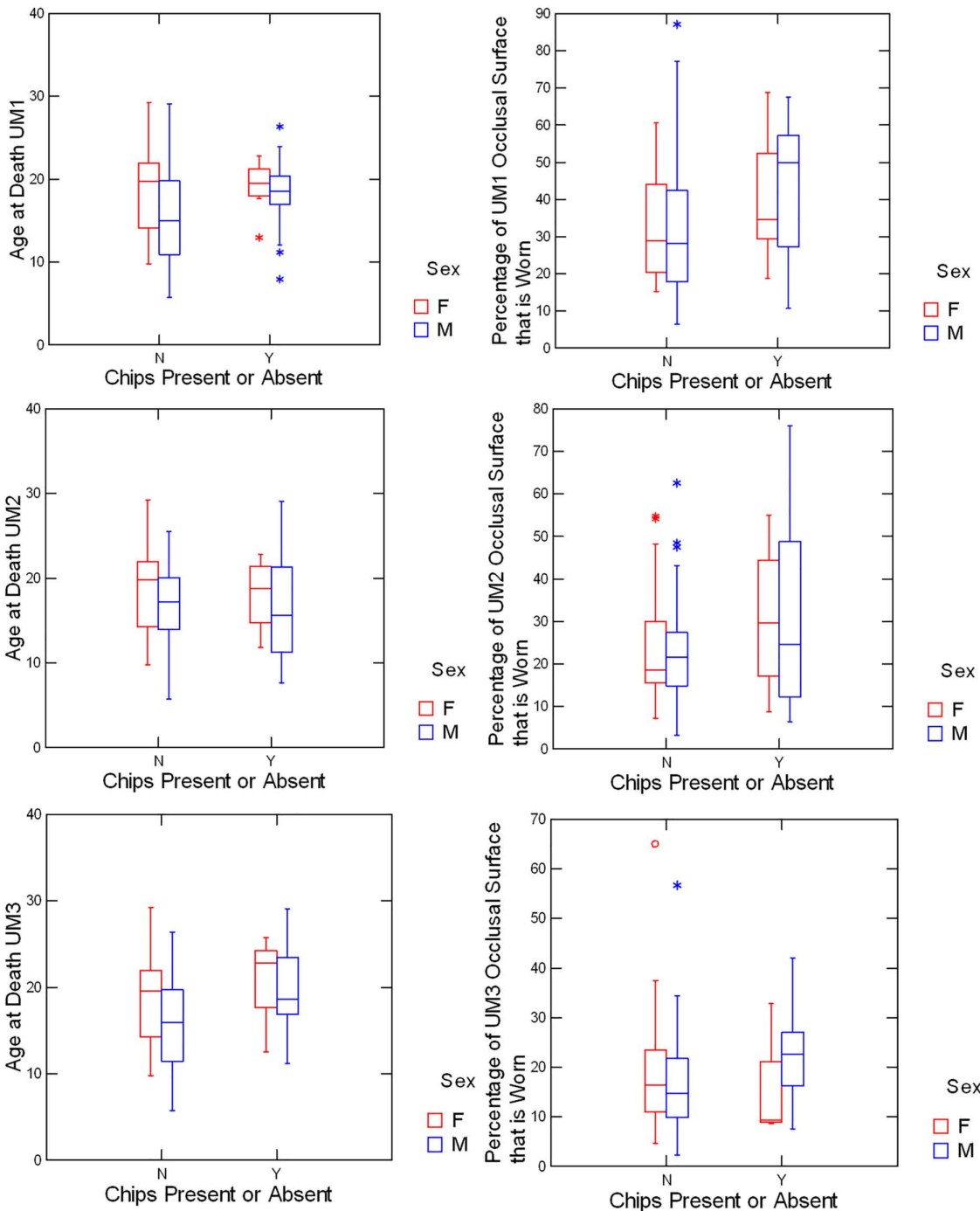

**Fig 3. Upper molars with and without enamel chips compared for age and wear.**

related more generally to crack formation [50]. We found evidence that this was the case. In our GLMM analysis, wear, not age, was a statistically significant predictor of chipping. The supplementary statistical analysis by tooth type also supports this finding, as wear was found to be a significant predictor for two tooth types, while age was a significant

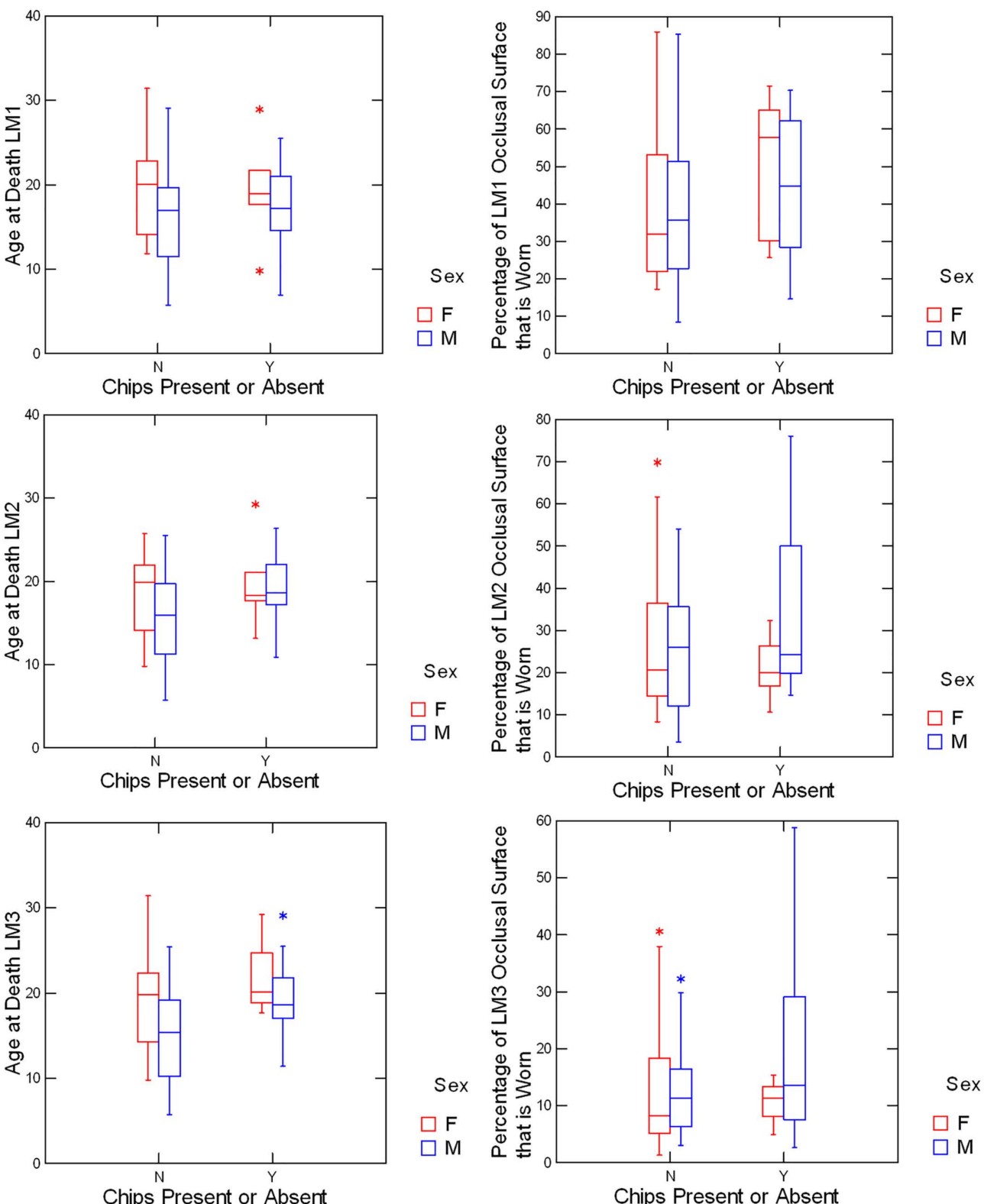

**Fig 4. Lower molars with and without enamel chips compared for age and wear.**

**Table 2. Spearman Rank Correlations between Percentage of Occlusal Surface that is Worn and Age-at-Death.**

| Molar Type | N | Spearman Correlation | p-value |
|---|---|---|---|
| UM1 | 98 | 0.698 | 0.001 |
| UM2 | 99 | 0.737 | 0.001 |
| UM3 | 98 | 0.321 | 0.001 |
| LM1 | 99 | 0.624 | 0.001 |
| LM2 | 101 | 0.763 | 0.001 |
| LM3 | 99 | 0.719 | 0.001 |

**Table 3. Results for GLMM repeated measures analysis of chipping as a function of various predictors: Type 3 Fixed Effects.**

| Effect | Numerator DF | Denominator DF | F value | p-value |
|---|---|---|---|---|
| Wear | 1 | 460 | 4.30 | 0.0387* |
| Tooth | 5 | 458 | 0.99 | 0.4248 |
| Sex | 1 | 92 | 7.15 | 0.0089* |
| Age-at-Death | 1 | 92 | 0.46 | 0.4973 |
| Year-of-Birth | 1 | 92 | 3.75 | 0.0559 |
| Time Spent in Sabana Seca | 1 | 92 | 0.21 | 0.6447 |

* = significant at p < 0.05.

**Table 4. Repeated Measures Analysis of Male vs. Female Occlusal Areas: Type 3 Fixed Effects.**

| Effect | Numerator DF | Denominator DF | F value | p-value |
|---|---|---|---|---|
| Molar | 5 | 99 | 427.35 | 0.0001 |
| Sex | 1 | 99 | 8.59 | 0.0042 |
| Molar*Sex | 5 | 99 | 2.69 | 0.0252 |

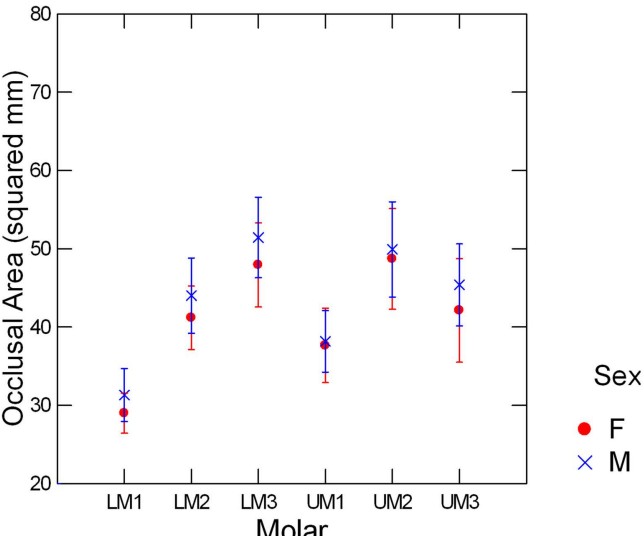

**Fig 5. Sex differences in occlusal areas.**

**Table 5. Chip areas and "h" in scallop-shaped edge chips with clear boundaries.**

| Molar Type | No. Females | No. Males | Mean female "h" (mm) (SD) | Mean male "h" (mm) (SD) | Mean female "h" scaled to occlusal area *100 (1/mm) (SD) | Mean male "h" scaled to occlusal area*100 (1/mm) (SD) |
|---|---|---|---|---|---|---|
| UM1 | 5 | 11 | 0.174 (0.082) | 0.221 (0.112) | 0.440 (0.164) | 0.563 (0.301) |
| UM2 | 1 | 7 | 0.077 | 0.378 (0.192) | 0.158 | 0.723 (0.394) |
| UM3 | 1 | 3 | 0.234 | 0.224 (0.062) | 0.484 | 0.494 (0.094) |
| LM1 | 5 | 9 | 0.330 (0.092) | 0.331 (0.067) | 1.124 (0.417) | 1.064 |
| LM2 | 4 | 4 | 0.416 (0.080) | 0.268 (0.071) | 0.977 (0.199) | 0.600 (0.172) |
| LM3 | 3 | 3 | 0.544 (0.192) | 0.425 (0.132) | 1.242 (0.500) | 1.15 (0.814) |
| Molar Type | No. Females | No. Males | Mean female Chip area (mm$^2$) (SD) | Mean male Chip area (mm$^2$) (SD) | Mean female Chip area scaled to occlusal area*100 | Mean male Chip area scaled to occlusal area*100 |
| UM1 | 5 | 11 | 0.064 (0.068) | 0.216 (0.277) | 0.159 (0.154) | 0.541 (0.685) |
| UM2 | 1 | 7 | 0.025 | 0.392 (0.298) | 0.051 | 0.747 (0.579) |
| UM3 | 1 | 3 | 0.146 | 0.218 (0.217) | 0.302 | 0.457 (0.414) |
| LM1 | 5 | 9 | 0.288 (0.177) | 0.361 (0.162) | 0.980 (0.640) | 1.172 (0.562) |
| LM2 | 4 | 4 | 0.414 (0.201) | 0.240 (0.208) | 1.690 (0.489) | 0.530 (0.457) |
| LM3 | 3 | 3 | 0.425 (0.132) | 0.330 (0.162) | 0.742 (0.164) | 0.611 (0.365) |

predictor for only one (the LM3). The tooth type results are discussed more fully in the "Summary and Discussion" section of the SI. Overall, our data suggest that chipping frequencies are affected by wear (and possibly also by age (LM3, tooth type analysis)), lending support to the view that wear can complicate reconstructions of fossil primate diets based on chipping frequencies [11,36].

It is notable that the time monkeys spent in Sabana Seca was not related to chipping in the GLMM analyses (nor in the tooth type analysis included in the SI). In their analysis of chipping frequencies across primates, Towle & Loch [18] found that terrestrial primate foragers tend to have higher chipping prevalence, suggesting that the mastication of grit may contribute to chip formation. This may also be the case for the Cayo Santiago rhesus monkeys, but the length of time these monkeys spent in the captive facility at Sabana Seca, where there is diminished opportunity for grit ingestion, does not reflect this possibility.

We also hypothesized, based on the tendency for human males to exhibit greater chipping frequences than human females [15], that if a sex difference were found, males would be more likely to have chipped molars than females. We found evidence that supports this hypothesis. Besides wear, sex was the only other statistically significant predictor in our GLMM. The effect of sex on chipping frequencies in our sample was independent of both wear and age (Table 3). Furthermore, inspection of Table 2 shows that males and females in this sample were quite similar in terms of dental wear, but not age; on average, males in this sample were three years younger than females. If the effect of sex on chipping were solely dependent on age, then males would not have been expected to have higher chipping frequencies than females.

Male Cayo Santiago monkeys tend to wear their teeth more rapidly than females do [70]. The similarity in dental wear for males and females overall in this sample is likely due to the three-year difference in their average ages. Thus, wearing their molars more rapidly, males had molars with a similar degree of wear to females that were on average three years older. Both the higher frequency of chipping in males as well as their faster rate of molar wear [70] counter the expectation that the sex difference in geophagic behavior in Cayo Santiago rhesus macaques could be a contributing factor to sex differences in chipping prevalence and dental wear rates. Being more geophagic [62], females might be expected to wear their teeth faster and to have higher frequencies of chipping, but this was not the case. It is worth noting that at Cayo Santiago, female rhesus macaques have earlier functional eruption of upper M1, and upper and lower M2s (around

4 months), yet later functional eruption of upper and lower M3s compared with male rhesus macaques, perhaps related to their different life history patterns and sexual maturity rates [76]. These eruption differences between the sexes may be a complicating factor here.

Unsurprisingly, males were generally found to have larger molars than females in this sample, potentially increasing the area for chipping [11,36]. We note, however, that UM2, one of the two molar types in which a sex difference in chipping frequency was found (the other being the LM3, see SI), is quite similar in size for the males and females of our sample (Fig 5). Thus, whether the sex difference in occlusal area might contribute to the sex difference in chipping frequencies is not clear from our data. What is also not clear is how cusps and regions where chips are likely to form scale with size in these teeth. Scaling differences could have an impact on male vs. female chipping rates.

We measured chip sizes because they often reflect bite force [1,10] and predicted that if males were applying larger bite forces, their chips would be larger than those of females. Our measurements of chip size—in terms of both "h" and chip areas—did not trend towards greater values in males. We therefore found no positive evidence from chip sizes that males are exerting higher bite forces than females. However, because our sample size for chip measurements was small, the possibility that male bite forces might result in larger chips (and potentially also explain their greater frequency of chipping) cannot be ruled out based on our results.

To summarize our analysis of the relationship between sex and enamel chipping: we conclude that the sex difference in chipping found in this study is not likely to be related to wear or age, but that sex differences in occlusal areas and bite force cannot be excluded as possible explanations. Another possible influence on sex differences in chipping frequencies, however, that we did not evaluate here is enamel thickness. If males have thinner enamel than females, then this might help to explain the greater probability of chipping in their teeth (as well as their tendency to wear their teeth at faster rates), although one study found that enamel thickness in macaques was not significantly different between males and females [43]. It is also possible that the molar occlusal surfaces of males might be subject to high forces during canine honing, potentially increasing both their chances of chipping and wear rates in comparison with those of females.

In their study of dental wear in Cayo Santiago rhesus macaques, Guatelli-Steinberg et al. [70] suggested that with larger body sizes to support, Cayo Santiago males might process greater quantities of food that wear their molars more rapidly. The possibility that body size might influence wear rates was indicated in the male sample of their study, for which there was a relationship between weight and dental wear, both measured in the living animals at the time of the 1985 roundup. (No such relationship was expected or found in the living female sample because adult female body weights fluctuate considerably with reproductive status). Chewing more than females, males might increase their chances of chipping their molars. If body weight differences between male and females influence their chipping frequencies, then, it is worth noting that macaques are characterized by "moderate to strong" body size dimorphism relative to other primates [77]. Thus, sex differences in chipping might be more apparent in primate species with moderate to strong body size and present less of a complication for paleoanthropological studies of chipping in species with minimal sexual dimorphism.

At an ultimate level of explanation, our sex-specific findings are consistent with the disposable soma hypothesis, which predicts that male investment in immediate growth and reproduction is at odds with long-term investments in dental tissue [78–80]. For sexually dimorphic males with limited reproductive lifespans, selection should favor the prioritization of rapid energetic intake and growth for intrasexual competition over behaviors that prolong tooth life [81]. Cayo Santiago rhesus macaque males wear their teeth faster than females [70], exhibit more dental pathologies and canine breaks [63], and chip their postcanine teeth more often (this study). Such findings are compatible with a 'wear fast and die young' strategy [81] that trades off long-term dental preservation for shorter-term energetic and reproductive returns. Based on previous work on food washing rates [80], we may expect enamel chipping to be rank-stratified as well as sex-stratified, if higher ranking individuals also encounter diminishing reproductive returns from extending their lifespans. This remains a priority for future research.

This study did not identify the most likely cause(s) of sex differences in chipping frequencies; yet it is noteworthy that such differences were found. As explained earlier, most studies of chipping that aim to detect a signal of hard-object feeding in fossilized teeth have not incorporated sex as a covariate. The present study suggests that sex, if it can be assessed from dental remains through the analysis of AMELX and AMELY proteins in teeth [38–42], might be important to consider in analyzing and interpreting chipping frequencies in fossil remains. In our study, depending on tooth type, males had up to three times higher chipping frequencies than females (see UM2, in Table 2). Primate species that eat hard foods generally have frequencies of enamel chipping (averaged over posterior teeth) that are greater than 25%, while those that do not eat them tend to have frequencies below 15% [18]. For females in our study, the average percentage of molars with one or more chips (across all six molar types) is 12.6%, while for males it is 22.8%. Although both frequencies are below the 25% threshold for hard-object feeders [18], females, at 12.6%, fall into the category of primates that do not tend to eat hard objects, while males, at 22.8%, exceed this value. Based on our findings, chipping frequency differences between males and females in one species can span a range of chipping frequencies among primate species with different diets, suggesting caution when analyzing chipping frequencies without reference to sex.

## Conclusion

In this study of chipping frequencies in Cayo Santiago rhesus macaques, we found evidence that wear is a stronger predictor of chipping frequencies than age, consistent with the direct link between wear and chip formation predicted by theoretical models. Here, we also found for the first time in a sample of modern primates, that males had higher frequencies of chipping than females. Sex differences in chipping frequencies as high as those found here could potentially obscure dietary signals gleaned from analyzing chipping frequencies in fossil primates. Our results suggest that dental wear and sex may be important factors to consider when using chipping frequencies to infer the diets of fossil primates.

## Supporting information

**S1 Text. Supplementary Information Analysis by Tooth Type.**
(DOCX)

**S2 Text. Metadata for Excel Spreadsheet.**
(DOCX)

**S3 Spreadsheet. Data for Study.**
(XLS)

**S4 Questionnaire. Inclusivity in Global Research.**
(DOCX)

## Acknowledgments

The Cayo Santiago macaque colony and derived skeletal collection housed at University of Puerto Rico Medical Center which are supported through the Cayo Santiago Primate Research Center by National Institutes of Health NIH contracts NIH 5P40OD012217. Dr. Melween I. Martinez Rodriguez, Dr. Carlos Sariol, Mr. Bonn V Liong Aure, Dr. Angelina Ruiz-Lambides, Dr. Alyssa Arre, Ms. Nahirí Rivera Barreto, and other CPRC staff members for their support and help. Special thanks are extended to the following CPRC staff members: Edgar Davila, Myrna Reyes and Alberto Clemente for performing skeletal preparations; and Mrs. Terry B. Kensler, Dr. Matthew Kessler, Dr. Rich Rawlins, Dr. Jean Turnquist, Mrs. Myriam Viñales, the late Dr. Nancy Hong, Dr. John Cant, Dr. Donald Dunbar, and Dr. Catalina Villamil for their curatorial work with or contributions to the skeletal collection. L.D.F. acknowledges support of a National Science Foundation Graduate Research Fellowship (GRFP award no. 1840344).

## Author contributions

**Conceptualization:** Debbie Guatelli-Steinberg, Luke D. Fannin.

**Data curation:** Debbie Guatelli-Steinberg, Luke D. Fannin.

**Formal analysis:** Debbie Guatelli-Steinberg.

**Funding acquisition:** Debbie Guatelli-Steinberg, Qian Wang.

**Investigation:** Debbie Guatelli-Steinberg, Luke D. Fannin.

**Methodology:** Debbie Guatelli-Steinberg, Luke D. Fannin.

**Project administration:** Debbie Guatelli-Steinberg, Qian Wang.

**Resources:** Qian Wang.

**Validation:** Debbie Guatelli-Steinberg.

**Writing – original draft:** Debbie Guatelli-Steinberg, Luke D. Fannin.

**Writing – review & editing:** Debbie Guatelli-Steinberg, Luke D. Fannin, Qian Wang.

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
