## [Decision Letter · Decision Letter 0]

15 Aug 2025

Dear Dr. Guatelli-Steinberg,

We look forward to receiving your revised manuscript.

Kind regards,

James J Cray Jr., Ph.D.

Academic Editor

PLOS ONE

Additional Editor Comments (if provided):

Reviewers' comments:

Reviewer's Responses to Questions

**Comments to the Author**

1. Is the manuscript technically sound, and do the data support the conclusions?

Reviewer #1: Partly

Reviewer #2: Yes

2. Has the statistical analysis been performed appropriately and rigorously?

Reviewer #1: No

Reviewer #2: Yes

3. Have the authors made all data underlying the findings in their manuscript fully available?

Reviewer #1: Yes

Reviewer #2: Yes

4. Is the manuscript presented in an intelligible fashion and written in standard English?

Reviewer #1: Yes

Reviewer #2: Yes

Reviewer #1: This manuscript presents a study of enamel chipping as it relates to wear, age, and sex in the Cayo Santiago rhesus macaque population. The article is well-written and the question is important for understanding dietary estimation in fossil taxa, and even for understanding the factors that might impact the evolution of tooth shape and enamel thickness among primates. That being said, the use of a large number of statistical analyses and inconsistent use of a Bonferroni-corrected significance threshold limit the utility of the findings. Please find my recommendations below:

Introduction

• Introduction doesn’t discuss the potential impact of enamel thickness on both wear and chipping, and how that might influence diet-wear-chipping relationships

o How does rhesus macaque enamel thickness vary and how might that influence the application of this study to other taxa?

o I appreciate that this is mentioned in the discussion, but incorporating some mention into the introduction would be helpful for understanding how macaques compare to the fossil primates we are hoping to better understand through this study of macaques

• The logic of lines 122-127 is unclear. Males have more missing/broken teeth in this population, but is that related to age or tooth wear? Why would that lead you to hypothesize that males will have more chipping than females even when wear is accounted for? Are there any specific feeding behaviors observed in males that would indicate a higher likelihood for chipping independently from wear and aging?

o Similarly, the next paragraph is a bit confusing. It is stated that there aren’t sex differences in food material properties, but then stated that females consume more soil than males. I would consider increased geophagy to be a sex difference in food material properties, but perhaps that’s not accurate?

• Line 139: Typo, “in in”

• Line 149: “Varying loads on brittle materials” seems to be missing a word.

• Line 177: “adaption” should be “adaptation”

Methods

• Line 178: “excel” should be capitalized

• It’s unclear why the multiple observations of tooth wear was described for these individuals since only the postmortem tooth wear seems to be used throughout the analyses… is this accurate?

• From my understanding of lines 211-222, it seems as though only half of the teeth were scored for wear (left or right), but chipping was scored on all teeth (left and right). Is that accurate? If so, that’s a big issue and if not, that should be clarified in the text.

• Lines 221-222: Blinding to age and sex makes sense, but it seems that by looking at the teeth the observed would be “unblinded” to wear, perhaps this could just be excluded?

• Line 241: “areas of were” may not need the “of”, or a word is missing

• Line 244: Why Spearman’s rank (rather than Pearson’s) unless variables are nonparametric? Normality test results could be reported if normality is a concern.

• Rather than running so many logistic regressions with Bonferroni, the authors should seriously consider using a GLMM with individual and tooth position as crossed random effects. Including all of the molars in one analysis would probably provide quite a bit more power.

o Both age and dental wear could also be included in a GLMM without running the Spearman’s rank assessment because imperfect collinearity does not violate the GLM’s assumptions

o This approach would seem to better address the question since the authors are not explicitly interested in tooth position, but in the frequency of chipping across the molar row

o This would also reduce the need for the Proc Mixed model between sexes… essentially you can just put all of the data into a single Proc Mixed model (or a series of models with model selection as described)

Results

• Unclear why maxillary and mandibular data are pooled for Mean Age at Death in Table 2 when the other variables are separated by tooth position, if the goal is just to demonstrate that females are older on average than males then why separate uppers and lowers?

• Reporting results as significant at two levels (with or without Bonferroni correction) somewhat defeats the purpose of the Bonferroni correction

o One of the benefits of the GLMM approach that I recommend is that Bonferroni correction would not be necessary

• Lines 366-367: Recommend specifying the variables (chip sizes) rather than saying, “In this table”, because what’s important is the finding rather than the table.

Discussion

• Line 376: This topic of crack formation independent of enamel chipping could be mentioned in the introduction since “crack” is used frequently to describe the formation of chips in the introduction of this paper

• Lines 399-401: Setting up the time in Sabana Seca as a natural experiment of the role of geophagy on enamel chipping is really clever and could be introduced in the introduction more clearly.

• Lines 409-412: The sex difference in age at death is not statistically significant (I’m pretty sure this wasn’t tested, but it doesn’t seem to be significant based on the SDs for the means), so this discussion of the impact of age differences by sex is probably overstating the impact of this difference, though the logic that this refutes the hypothesis that geophagy is contributing to higher rates of enamel chipping is still well-founded

• Lines 427-432: How would increased area of the tooth relate to scaling of the cusps and regions where chips are likely to form based on the mechanics of enamel fracturing? This could merit more discussion.

• Could the sex differences in chipping be related to different metabolic requirements in males and females? Are males simply chewing more than females? There is considerable focus on the finding of sex differences in chipping frequency, and it may be worth mentioning that male and female rhesus macaques are more sexually dimorphic than many fossil primates to which we might be interested in applying these results

o The differences between the sexes could even be thought to invalidate the 15% and 25% thresholds for hard object feeding since these males and females have such similar diets and fall into different categories. Perhaps these thresholds need to better account for not only sex but look at species-specific patterns of chipping?

Reviewer #2: This manuscript reports on a study of the relationship of molar wear, age and sex to enamel chipping in a sample of rhesus macaques of known age from Cayo Santiago. The authors discuss the importance of this relationship for diet and food ecology of fossil primates. The authors extrapolate these results to the broader considerations of dental tissue growth and development and reproductive investment.

The paper is well written and well organized. The literature review is good and appropriate, and the specific aims are clearly stated. Methods are clear and appropriate, as are the results. The discussion clearly presents the results as they relate to the specific aims and the application of these results to the study of fossil primates.

Figures and tables are clearly presented.

I have minor revisions to suggest. I do not need to review this manuscript again unless requested by the corresponding editor:

Abstract

Line 39: The Bonferroni level of significance is unclear. Making this clearer where the report is of one tooth (with P = 0.05) and multiple teeth (with the Bonferroni level of significance) would be helpful

Methods

Table 1: Should the P-value for each of the results should be indicated as < 0.001? It seems unlikely that they are all equal to 0.001.

lines 285-286: Clarify that this is a regression, not a mixed-model ANOVA.

Results

line 307: should the second use of “chipping” be removed?

Table 3: Table legend should include an indication of * and **. You also might consider using something other than “*” and “**” since the * is used in Table 5 for another meaning. This could simplify the presentation for other readers.

Discussion

1st paragraph (lines 374-379). This is a lot of information in 2 sentences. The first sentence could be broken up to present the results more clearly.

Figure 2 legend – The wording is awkward. Is a word missing? This is more clearly stated on line 205.

**Do you want your identity to be public for this peer review?** For information about this choice, including consent withdrawal, please see our Privacy Policy

Reviewer #1: No

Reviewer #2: No

---

## [Author Response · Author response to Decision Letter 1]

22 Sep 2025

Response to Reviewers

We thank the reviewers for the effort and thoughtful comments and suggestions, which we respond to below.

Reviewer #1: This manuscript presents a study of enamel chipping as it relates to wear, age, and sex in the Cayo Santiago rhesus macaque population. The article is well-written and the question is important for understanding dietary estimation in fossil taxa, and even for understanding the factors that might impact the evolution of tooth shape and enamel thickness among primates. That being said, the use of a large number of statistical analyses and inconsistent use of a Bonferroni-corrected significance threshold limit the utility of the findings. Please find my recommendations below:

Response: As we explain in detail below, based on Reviewer 1’s recommendation, we replaced the original set of by-tooth logistic regression analyses in the main text (Table 3) with a new repeated measures GLMM analysis in which values of variables for each tooth type are treated as repeated measures for each individual. Importantly, the new results from this GLMM are consistent with our original analyses, namely in finding statistically significant effects on chipping frequencies for both wear and sex. As noted by Reviewer 1, the advantage of the new analysis is that it has higher power than our previous analyses.

We include the original analyses in the Supplementary Information because it allows us to: (1) provide a “by-tooth” assessment of the effects of predictors that might be of interest to other researchers, and (2) simultaneously eliminate any concerns regarding the collinearity of wear and age.

Schielzeth et al. (2020) [Schielzeth, H., Dingemanse, N.J., Nakagawa, S., Westneat, D.F., Allegue, H., Teplitsky, C., Réale, D., Dochtermann, N.A., Garamszegi, L.Z. and Araya‐Ajoy, Y.G., 2020. Robustness of linear mixed‐effects models to violations of distributional assumptions. Methods in Ecology and Evolution, 11(9), pp.1141-1152] explain that, as Reviewer 1 pointed out, linear mixed models are robust to collinear predictors. The authors of this paper found that weak to moderate correlations had little effect on the precision of GLMM estimates but that there were large effects on the precision of estimates for predictors with correlations as high as 0.8. Correlations for wear and age do not reach this level in our data but for some tooth types, they are close (Table 2, main text). Thus, as noted above, we include our original analysis in the SI, in part because it eliminates any concern regarding collinearity of these variables.

Although Reviewer 1 found our use of Bonferroni correction to be inconsistent, we believe that we used the correction consistently but reported the traditional level of statistical significance as well. We did this because there is longstanding and continuing controversy over when and how to use the Bonferroni correction and other multiple correction methods. Some researchers, including statisticians, find the Bonferroni correction to be overly conservative and even question whether corrections for multiple testing are necessary (e.g., Cabin, R.J. and Mitchell, R.J., 2000. To Bonferroni or not to Bonferroni: when and how are the questions. Bulletin of the Ecological Society of America, 81(3), pp.246-248.201]; Streiner, D.L. and Norman, G.R., 2011 Correction for multiple testing: is there a resolution? Chest, 140(1), pp.16-18]; and Armstrong, R.A., 2014. When to use the Bonferroni correction. Ophthalmic and Physiological Optics, 34(5), pp.502-508). Armstrong (2014) does not recommend the use of the Bonferroni correction except for situations in which it is “imperative to avoid Type I error” and when “…a large number of tests are carried out without preplanned hypotheses in an attempt to establish any results that may be significant” (Armstrong 2014:505). In further considering the need for a Bonferroni or other multiple testing correction in our supplementary analysis, we realize that our study does not satisfy either of Armstrong’s (2014) conditions. We now include this reasoning in the Supplementary Information document labelled “Supplementary Information: Analysis by Tooth Type.”

Reviewer #1: Introduction doesn’t discuss the potential impact of enamel thickness on both wear and chipping, and how that might influence diet-wear-chipping relationships

o How does rhesus macaque enamel thickness vary and how might that influence the application of this study to other taxa?

o I appreciate that this is mentioned in the discussion, but incorporating some mention into the introduction would be helpful for understanding how macaques compare to the fossil primates we are hoping to better understand through this study of macaques.

Response: We appreciate this suggestion. We now include a new paragraph in the introduction of the revised MS discussing macaques as a model for understanding how “non-food” factors --such as dental wear, age, and sex-- might affect enamel chipping frequencies in fossil primates (including hominins). This paragraph includes an evaluation of rhesus macaque enamel thickness and molar sexual dimorphism.

Reviewer #1: The logic of lines 122-127 is unclear. Males have more missing/broken teeth in this population, but is that related to age or tooth wear? Why would that lead you to hypothesize that males will have more chipping than females even when wear is accounted for? Are there any specific feeding behaviors observed in males that would indicate a higher likelihood for chipping independently from wear and aging?

Response: We appreciate the reviewer’s point about the unclear logic linking broken teeth (in this case canines) to molar enamel chipping. However, we also mentioned that in humans there is a tendency for males to have higher frequencies of chipping. That was actually the second “empirical finding” that influenced our thinking. We have rephrased our expectation of higher frequencies of chipping in Cayo Santiago males, basing it solely on the finding that in humans, when sex differences in chipping have been found, males generally have higher chipping frequencies than females. It is not clear why this is the case, though it could have to do with sex differences in bite force, tooth grinding, quantity of food ingested, the material properties of foods, and/or sex differences in enamel thickness. We also removed the words “when age and wear were accounted for” since it is not clear why, in humans, males tend to have greater frequencies of chipping than females do. Finally, we do not know of any specific feeding behaviors that Cayo Santiago rhesus males engage in that would indicate a higher likelihood of chipping.

Reviewer #1: Similarly, the next paragraph is a bit confusing. It is stated that there aren’t sex differences in food material properties, but then stated that females consume more soil than males. I would consider increased geophagy to be a sex difference in food material properties, but perhaps that’s not accurate?

Response: We see Reviewer #1’s point, but it isn’t clear to us that soil is always a “food.” Primates seem to consume soil for a variety of reasons, including as a method of self-medication (Krishnamani and Mahaney, 2000: Krishnamani, Ramanathan, and William C. Mahaney. "Geophagy among primates: adaptive significance and ecological consequences." Animal Behaviour 59, no. 5 (2000): 899-915). For this reason, we prefer not to presume that soil is eaten as a food (i.e., similar to monkey chow or other substance consumed for nourishment).

Reviewer #1: Corrections

• Line 139: Typo, “in in”

• Line 149: “Varying loads on brittle materials” seems to be missing a word.

• Line 177: “adaption” should be “adaptation”

Methods

• Line 178: “excel” should be capitalized

Response: These errors have been corrected.

Reviewer #1: It’s unclear why the multiple observations of tooth wear was described for these individuals since only the postmortem tooth wear seems to be used throughout the analyses… is this accurate?

Response: Sorry for our lack of clarity. We have rewritten this sentence to be clear that we used the postmortem tooth wear measurements: “The sample and dental wear measurements made on the deceased monkeys that were used for that study are the same as those used here. “

Reviewer #1: From my understanding of lines 211-222, it seems as though only half of the teeth were scored for wear (left or right), but chipping was scored on all teeth (left and right). Is that accurate? If so, that’s a big issue and if not, that should be clarified in the text.

Response: Again, sorry for our lack of clarity. All teeth were originally scored for both wear and chipping. Then, the most-well oriented antimere was used in our analysis for both wear and chipping. We have clarified this in the text: “For our analyses, for each antimeric pair, we used the most-well oriented antimere for both wear and chipping.”

Reviewer #1: • Lines 221-222: Blinding to age and sex makes sense, but it seems that by looking at the teeth the observed would be “unblinded” to wear, perhaps this could just be excluded?

Response: That is a good point: we have removed this sentence.

Reviewer #1: Line 241: “areas of were” may not need the “of”, or a word is missing

Response: Corrected: removed word “of”

Reviewer #1: Line 244: Why Spearman’s rank (rather than Pearson’s) unless variables are nonparametric? Normality test results could be reported if normality is a concern.

Response: We had indeed conducted normality tests but did not include them. We do so now in the Materials and Methods section. We used Spearman’s rank rather than Pearson because Shapiro-Wilk tests revealed that the variable “wear” was not normal for any of the six tooth types.

Reviewer #1: Rather than running so many logistic regressions with Bonferroni, the authors should seriously consider using a GLMM with individual and tooth position as crossed random effects. Including all of the molars in one analysis would probably provide quite a bit more power.

o Both age and dental wear could also be included in a GLMM without running the Spearman’s rank assessment because imperfect collinearity does not violate the GLM’s assumptions

o This approach would seem to better address the question since the authors are not explicitly interested in tooth position, but in the frequency of chipping across the molar row

o This would also reduce the need for the Proc Mixed model between sexes… essentially you can just put all of the data into a single Proc Mixed model (or a series of models with model selection as described)

Response: We recognize that imperfect collinearity does not violate GLMM assumptions as explained by: Schielzeth, H., Dingemanse, N.J., Nakagawa, S., Westneat, D.F., Allegue, H., Teplitsky, C., Réale, D., Dochtermann, N.A., Garamszegi, L.Z. and Araya‐Ajoy, Y.G., 2020. Robustness of linear mixed‐effects models to violations of distributional assumptions. Methods in Ecology and Evolution, 11(9), pp.1141-1152. The authors of this paper found that weak to moderate correlations had little effect on the precisions of GLMM estimates but that there were large effects on the precision of estimates for predictors with correlations as high as 0.8. Correlations for wear and age in our data set do not reach this level but for some tooth types, they are close. As noted above, our original analysis, now included in the SI, eliminates any concern regarding collinearity of these variables, simultaneously providing analyses by tooth type. Our original analysis is consistent with our new GLMM analysis in finding significant effects for wear and age. In our original analysis, wear and sex were the most frequent statistically significant predictors across tooth types. In our new analysis, wear and age are the only statistically significant predictors.

In our GLMM, we designated “individual” as random but tooth type as a fixed effect. We believe that tooth type may have a consistent effect on the likelihood of chipping, as for example, third molars erupt later than second molars and tend to have thicker enamel than first and second molars, at least in humans (Smith, T.M., Olejniczak, A.J., Reid, D.J., Ferrell, R.J. and Hublin, J.J., 2006. Modern human molar enamel thickness and enamel–dentine junction shape. Archives of Oral Biology, 51(11), pp.974-995.) Both of these factors would tend to result in systematically lower frequencies of chipping in third molars. These two factors could explain why the three tooth types in our sample with the lowest frequencies of chipping were all third molars (although our GLMM ultimately did not reveal significant effects by tooth type). Our new GLMM analysis is fully described on page in the statistical methods section of the revised manuscript. Our results are incorporated into a new Table, labeled Table 3 of the revised manuscript.

Reviewer 1: Results: Unclear why maxillary and mandibular data are pooled for Mean Age at Death in Table 2 when the other variables are separated by tooth position, if the goal is just to demonstrate that females are older on average than males then why separate uppers and lowers?

Response: We have revised the Table (which is now Table 1) to report the age of individuals rather than separating by upper and lower dentitions.

Reviewer 1: Reporting results as significant at two levels (with or without Bonferroni correction) somewhat defeats the purpose of the Bonferroni correction

Response: Please see our response above re: Bonferroni correction and why we no longer apply it.

Reviewer 1: One of the benefits of the GLMM approach that I recommend is that Bonferroni correction would not be necessary

Response: Yes—this was a much-appreciated suggestion. Please see our response above for how we carried out our GLMM analysis.

Reviewer 1: Lines 366-367: Recommend specifying the variables (chip sizes) rather than saying, “In this table”, because what’s important is the finding rather than the table.

Response: Corrected.

Reviewer 1: Discussion: Line 376: This topic of crack formation independent of enamel chipping could be mentioned in the introduction since “crack” is used frequently to describe the formation of chips in the introduction of this paper

Response: To be more clear ( hopefully) when we first use the word “crack” in the introduction we added the words “in some instance” before “creating a chip,” as follows: “Biting objects with the force necessary to fracture them can cause enamel, a brittle material, to crack, in some instances creating a chip (or more precisely a chipping scar) on the enamel surface [1].” In in our second use of the word “crack” in the introduction, we now include in parentheses, after the word crack: “i.e., fractures within the enamel.”

Reviewer 1: Lines 399-401: Setting up the time in Sabana Seca as a natural experiment of the role of geophagy on enamel chipping is really clever and could be introduced in the introduction more clearly.

Response: We thank Reviewer 1 for this excellent suggestion. We now include expectations regarding the role of geophagy (and the comparison between free-ranging and captive conditions) on enamel chipping in the introduction.

Reviewer 1: Lines 409-412: The sex difference in age at death is not statistically significant (I’m pretty sure this wasn’t tested, but it doesn’t seem to be significant based on the SDs for the means), so this discussion of the impact of age differences by sex is probably overstating the impact of this difference, though the logic that this refutes the hypothesis that geophagy is contributing to higher rates of enamel chipping is still well-founded

Response:

We softened the language here, in attempt not to overstate this point, now stated as “Also evident from Table 2, males in this sample were three years younger than females, on average. If the sex effect were solely dependent on age, then males would not have been expected to have higher chipping frequencies than females.”

Reviewer 1: Lines 427-432: How would increased area of the tooth relate to scaling of the cusps and regions where chips are likely to form based on the mechanics of enamel fracturing? This could merit more discussion.

Response: Agreed, but we do not know how cusps and regions where chip

---

## [Decision Letter · Decision Letter 1]

4 Nov 2025

Dear Dr. Guatelli-Steinberg,

We look forward to receiving your revised manuscript.

Kind regards,

James J Cray Jr., Ph.D.

Academic Editor

PLOS ONE

Journal Requirements:

Additional Editor Comments:

There are two very minor issues to be addressed prior to acceptance.

Reviewers' comments:

Reviewer's Responses to Questions

**Comments to the Author**

Reviewer #1: (No Response)

Reviewer #2: (No Response)

2. Is the manuscript technically sound, and do the data support the conclusions?

Reviewer #1: Yes

Reviewer #2: Yes

3. Has the statistical analysis been performed appropriately and rigorously?

Reviewer #1: Yes

Reviewer #2: Yes

4. Have the authors made all data underlying the findings in their manuscript fully available?

Reviewer #1: Yes

Reviewer #2: Yes

5. Is the manuscript presented in an intelligible fashion and written in standard English?

Reviewer #1: Yes

Reviewer #2: Yes

Reviewer #1: The authors have done an excellent job of addressing my concerns. I have only a small number of minor notes on this version of the manuscript:

Page 12, line 257: “chip occlusal areas of” seems to have an unnecessary “of”

Page 14, line 294: “are” should be “as” in this line

Page 18: p-values throughout are listed as “p < number” when they may be better listed as “p = number”

Page 25, lines 471-473: This sentence seems to have a misplaced comma

Reviewer #2: This paper is well written, and the authors have incorporated the reviewers’ comments thoroughly. The revised analyses and results are explained in a clear manner. I appreciate that the authors included the original results in the supplemental material for those interested in the analysis of individual teeth. My original comments were well addressed.

I have only two minor comments, both related to figure legends.

Figure 1 legend: the c figure is included twice. Should these descriptions be combined? Also, the colors used in the figure are quite similar. Perhaps either colors that contrasting more, or include one of the colors by dotted lines for individuals with color blindness.

Figure 2 legend – The first part of this figure legend is awkward. It seems that there is either a word missing, or extra words.

**Do you want your identity to be public for this peer review?** For information about this choice, including consent withdrawal, please see our Privacy Policy

Reviewer #1: No

Reviewer #2: No

---

## [Author Response · Author response to Decision Letter 2]

5 Nov 2025

Response to Editor

Thank you for your careful review of our paper. Reviewers had minor revisions, which we have addressed in the revised MS. Below, we detail the changes that were made in this version of our MS.

REVIEWER 1: The authors have done an excellent job of addressing my concerns. I have only a small number of minor notes on this version of the manuscript:

Page 12, line 257: “chip occlusal areas of” seems to have an unnecessary “of”

Page 14, line 294: “are” should be “as” in this line

RESPONSE: These two items have been corrected

REVIEWER 1: Page 18: p-values throughout are listed as “p < number” when they may be better listed as “p = number”

RESPONSE: We originally reported p-values with “=” signs as they are exact p-values. Reviewer 2 suggested we change them to inequalities. Reviewer 1 (as noted above), however, states that these should be listed as “p=number” (which is what we thought originally). We have looked further into this issue, and it seems Reviewer 1 is correct: these should be reported as exact p-values with an equal sign. We have made this change throughout the MS.

RESPONSE: These two items have been corrected

REVIEWER 1: Page 25, lines 471-473: This sentence seems to have a misplaced comma

RESPONSE: Corrected.

REVIEWER #2: This paper is well written, and the authors have incorporated the reviewers’ comments thoroughly. The revised analyses and results are explained in a clear manner. I appreciate that the authors included the original results in the supplemental material for those interested in the analysis of individual teeth. My original comments were well addressed. I have only two minor comments, both related to figure legends.

Figure 1 legend: the c figure is included twice. Should these descriptions be combined? Also, the colors used in the figure are quite similar. Perhaps either colors that contrasting more, or include one of the colors by dotted lines for individuals with color blindness.

RESPONSE: We are showing something different in “c” so we prefer to leave “c” as it is. We agree that the green and blue in this figure would be problematic for individuals with blue-yellow color blindness. We have changed the colors in Figure 1 to blue/red.

REVIEWER #2: Figure 2 legend – The first part of this figure legend is awkward. It seems that there is either a word missing, or extra words.

RESPONSE: Sorry, we are not seeing where the reviewer finds the wording to be awkward. For this reason, we have not changed this figure legend.

Many thanks for your time and for helping us improve this MS.

Best wishes,

Debbie Guatelli-Steinberg and co-authors

---

## [Editor Report · Decision Letter 2]

11 Nov 2025

How wear, age, and sex relate to enamel chipping in Cayo Santiago rhesus macaques (Macaca mulatta)

PONE-D-25-35913R2

Dear Dr. Guatelli-Steinberg,

We’re pleased to inform you that your manuscript has been judged scientifically suitable for publication and will be formally accepted for publication once it meets all outstanding technical requirements.

Kind regards,

James J Cray Jr., Ph.D.

Academic Editor

PLOS ONE
---

## [Editor Report · Acceptance letter]

PONE-D-25-35913R2

PLOS ONE

Dear Dr. Guatelli-Steinberg,

I'm pleased to inform you that your manuscript has been deemed suitable for publication in PLOS ONE. Congratulations! Your manuscript is now being handed over to our production team.

Kind regards,

on behalf of

Dr. James J Cray Jr.

Academic Editor

PLOS ONE